# Metabolic Defects Caused by High-Fat Diet Modify Disease Risk through Inflammatory and Amyloidogenic Pathways in a Mouse Model of Alzheimer’s Disease

**DOI:** 10.3390/nu12102977

**Published:** 2020-09-29

**Authors:** Austin M. Reilly, Andy P. Tsai, Peter B. Lin, Aaron C. Ericsson, Adrian L. Oblak, Hongxia Ren

**Affiliations:** 1Stark Neurosciences Research Institute, Medical Neuroscience Graduate Program, Indiana University School of Medicine, Indianapolis, IN 46202, USA; aureilly@iu.edu (A.M.R.); tandy@iu.edu (A.P.T.); pblin@iu.edu (P.B.L.); aoblak@iupui.edu (A.L.O.); 2Metagenomics Center, University of Missouri, Columbia, MO 65201, USA; EricssonA@missouri.edu; 3Herman B. Wells Center for Pediatric Research, Department of Pediatrics, Indiana University School of Medicine, Indianapolis, IN 46202, USA; 4Center for Diabetes and Metabolic Diseases, Indiana University School of Medicine, Indianapolis, IN 46202, USA; 5Department of Biochemistry & Molecular Biology, Indiana University School of Medicine, Indianapolis, IN 46202, USA; 6Department of Pharmacology & Toxicology, Indiana University School of Medicine, Indianapolis, IN 46202, USA; 7Department of Anatomy and Cell Biology, Indiana University School of Medicine, Indianapolis, IN 46202, USA

**Keywords:** diet, metabolism, nutrient, glucose, lipid, insulin, neuroinflammation, Alzheimer’s disease

## Abstract

High-fat diet (HFD) has been shown to accelerate Alzheimer’s disease (AD) pathology, but the exact molecular and cellular mechanisms remain incompletely understood. Moreover, it is unknown whether AD mice are more susceptible to HFD-induced metabolic dysfunctions. To address these questions, we used 5xFAD mice as an Alzheimer’s disease model to study the physiological and molecular underpinning between HFD-induced metabolic defects and AD pathology. We systematically profiled the metabolic parameters, the gut microbiome composition, and hippocampal gene expression in 5xFAD and wild type (WT) mice fed normal chow diet and HFD. HFD feeding impaired energy metabolism in male 5xFAD mice, leading to increased locomotor activity, energy expenditure, and food intake. 5xFAD mice on HFD had elevated circulating lipids and worsened glucose intolerance. HFD caused profound changes in gut microbiome compositions, though no difference between genotype was detected. We measured hippocampal mRNAs related to AD neuropathology and neuroinflammation and showed that HFD elevated the expression of apoptotic, microglial, and amyloidogenic genes in 5xFAD mice. Pathway analysis revealed that differentially regulated genes were involved in insulin signaling, cytokine signaling, cellular stress, and neurotransmission. Collectively, our results showed that 5xFAD mice were more susceptible to HFD-induced metabolic dysregulation and suggest that targeting metabolic dysfunctions can ameliorate AD symptoms via effects on insulin signaling and neuroinflammation in the hippocampus.

## 1. Introduction

Alzheimer’s disease (AD) is the most prevalent cause of dementia in the elderly. AD is a progressive and devastating neurological disease, which begins with mild memory loss and eventually can seriously compromise a person’s ability to carry out daily activities. With its increasing prevalence in today’s aging society, AD has become a pressing global health concern [1]. The most characteristic neuropathological hallmarks of AD are the accumulation of extracellular Aβ plaques and intracellular neurofibrillary Tau tangles in the brain [2]. Additional AD biomarkers may lead to earlier diagnosis and interventions targeting the preclinical, asymptomatic stages of the disease [3]. Key molecular pathways have been implicated in the initiation and progression of the neuropathological cascade, which could provide potential targets for developing biomarkers and therapeutic strategies. A growing body of recent studies shows that increased neuroinflammation and impaired cellular metabolism may be the underlying cause of AD pathology [4,5,6,7]. Compelling epidemiological evidence points to a mechanistic connection between impaired metabolic homeostasis, age-associated cognitive impairment, and neurodegenerative diseases characterized by cognitive disorders [8,9]. Patients with diabetes develop more cognitive dysfunction and have a greater incidence of AD than non-diabetics [8]. In a meta-analysis of prospective studies, diabetes increased the relative risk of AD by 56% [10]. Glycated hemoglobin (HbA1c), a biomarker of diabetes severity and duration, is a top correlate of brain atrophy [11]. Aberrant glucose metabolism has been linked to amyloid deposition and brain cognitive dysfunction [12,13,14]. Brain insulin resistance causes metabolic and bioenergetic defects, which is a potential contributing factor to cognitive impairment and AD pathogenesis [9,15,16,17]. With the prevalence of both AD and diabetes on the rise, studies of potential mechanisms that underlie common predisposing factors are paramount.

Studies have suggested that diet and nutrition may play a role in the development of AD [18]. The Mediterranean diet (MeDi) is associated with a lower incidence of chronic diseases and shows protective effects against cognitive decline in aging individuals [19,20,21,22]. Conversely, the overconsumption of high-sugar and high-fat diets coupled with sedentary lifestyle predisposes individuals to metabolic diseases and neurocognitive defects during the aging process. High-fat diet (HFD) causes nutritional excess and promotes obesity and other key components of metabolic syndrome, such as systemic inflammation, dyslipidemia, insulin resistance, and elevated blood glucose [23] thereby contributing to AD pathogenesis. Indeed, previous studies showed that HFD feeding in a transgenic mouse model harboring five familial AD mutations (5xFAD) had increased amyloid deposition and impaired performance in memory and learning tasks [24,25]. 5xFAD is an early onset mouse model of Alzheimer’s disease harboring five AD-associated mutations in human *APP* and *PS1* which causes rapid progression of amyloid pathology due to the increased generation of insoluble Aβ isoforms [26]. However, the mechanisms linking high-fat diet to AD progression are still under investigation, and it remains unknown whether transgenic AD mouse models are more susceptible to HFD-induced metabolic derangements. Moreover, the exact molecular mechanisms of how nutrient excess caused by HFD feeding exacerbates AD pathophysiology remain largely undefined. In the present study, we set out to address these questions by studying the metabolic phenotype in the 5xFAD mouse model on normal chow diet (NCD) and high-fat diet (HFD). Our results demonstrated that HFD more severely impacted the metabolic homeostasis of transgenic AD mice compared to control mice. Furthermore, we characterized the changes in the gut microbiome and profiled hippocampal gene transcription of transgenic AD mice on NCD versus HFD. We revealed the specific effects of HFD on the hippocampi of the transgenic AD mice in terms of individual gene transcription and the collective changes in the neuropathology and neuroinflammatory pathways, thereby providing potential targets for AD therapy.

## 2. Experimental Procedures

### 2.1. Experimental Animals

All mice were maintained in the Indiana University School of Medicine Lab animal resource center (LARC) facility on a 12:12 h light: dark cycle. All animal protocols were approved by Indiana University IACUC (IACUC#11121, 11258, 19013, 20007, PI’s Ren and Oblak).). 5xFAD (mouse line Tg6799) heterozygous mice [26] are available from The Jackson Laboratory (https://www.jax.org/strain/008730) and a colony is maintained at Indiana University. Age-matched littermates lacking 5xFAD, hereafter called wild type (WT) mice, were used as controls. For gene expression studies, NCD mice were anesthetized with Avertin and transcardially perfused with ice-cold PBS. Frozen brain tissue was homogenized in T-PER buffer, RNase-free water, and stored in an equal volume of STAT-60 (tel-test inc, CS502) at −80 °C. HFD mice were euthanized with CO_2_ and tissues were stored at −80 °C.

### 2.2. Dietary Treatment

For metabolic profiling studies presented in Figure 1, Figure 2, Figure 3 and Figure 4 and Appendix A, mice were raised on 62.1% of calories from carbohydrates, 24.6% from protein, and 13.2% from fat (LabDiet, catalog #5053, Richmond, IN, USA) prior to starting HFD (Appendix A). At 3 months old, mice were switched to HFD containing 60% calories from lard-based fat, 20% from protein, 20% from carbohydrate (Research Diets, catalog #D12492, New Brunswick, NJ, USA). Mice had ad libitum access to food except for fasting/refeeding experiments as indicated in the figures and legends. Lean mass and fat mass were determined by MRI scan (EchoMRI-100, EchoMRI Houston, TX, USA). For microbiome and gene expression profiling studies presented in Figure 5 and Figure 6 and (Appendix A), age-matched NCD-fed WT and 5xFAD mice were fed 19.3% protein, 16.6% fat, 61.3% carbohydrates (LabDiet, catalog #5K52, Richmond, IN, USA) (Appendix A).

### 2.3. Glucose Measurements

Tail blood glucose was measured with AlphaTRAK 2 (Zoetis Inc., catalog #71681-01 and 71676-01, Kalamazoo, MO, USA) during ad libitum feeding, after 5 h of daytime fasting, or after 16 h overnight fasting. Oral glucose tolerance test (oGTT, 2 g/kg) in NCD-fed mice was performed after 16-h fasting. HFD-fed mice were given oGTT (3 g/kg) after 5-h fasting.

### 2.4. Serum Biochemistries

Serum was collected by tail vein bleeding or cardiac puncture during ad libitum feeding, daytime short fasting (5 h), overnight fasting (~16 h), and refeeding (4–5 h of feeding after overnight fast). Serum insulin was measured by ELISA (EMD Millipore, catalog #EZRMI-13K, Bellerica, MA, USA). Colorimetric assays were used to detect serum triglycerides (Thermo Fisher, catalog #TR22421, Middletown, VA, USA), free cholesterol E (Wako, catalog #990-02511, Chuo-Ku Osaka, Japan), glycerol (Sigma, catalog #F6428-40mL, St. Louis, MO, USA), and non-esterified fatty acids (NEFA) (Wako, catalog #999-34691, 995-34791, 991-34891, 993-35191, Chuo-Ku Osaka, Japan). All reactions were performed according to manufacturer protocols.

### 2.5. Indirect Calorimetry

Indirect calorimetry measurements were collected using a TSE PhenoMaster Platform (TSE Systems, Chesterfield, MO, USA) as described previously [27]. Briefly, mice were individually housed for a 48 h acclimation period before recording data used for analysis. Metabolic parameters (locomotor activity, food intake, energy expenditure, oxygen consumption, respiratory exchange ratio) were measured at 36 min intervals during a normal 12-h light/dark cycle. Total body weight and lean mass were determined beforehand by MRI scan (EchoMRI-100, EchoMRI Houston, TX, USA) for calculations. 

### 2.6. Hippocampal mRNA Quantitation

For each mouse, RNA was extracted from one hippocampus. HFD samples (left or right hippocampi) were extracted using Trizol reagent (Invitrogen, catalog #15596018, Carlsbad, CA, USA). NCD samples (left hippocampus) were extracted using STAT-60 reagent (Tel-Test, catalog #CS-502, Friendswood, TX, USA) and purified by using the Purelink RNA Mini Kit (Life Technologies, catalog #12183025, Carlsbad, CA, USA). The mRNA transcripts were detected using sequence-specific fluorescently barcoded probes (Nanostring Technologies, nCounter Neuropathology and nCounter Neuroinflammation; catalog numbers XT-CSO-MNROP1-12 and XT-CSO-MNROI1-12, respectively, Seattle, WA, USA). 200 ng of RNA was loaded for all samples and hybridized with probes for 16 h at 65 degrees Celsius. Results obtained from nCounter MAX Analysis System (NanoString Technologies, catalog #NCT-SYST-LS, Seattle WA) were imported to nSolver Analysis Software (v4.0; NanoString Technologies) for QC verification, normalization, and data statistics using Advanced Analysis (v2.0.115; NanoString Technologies). Probes were only included if the read count was more than 3 standard deviations above background, and probes that had <100 reads for 6 or more samples were removed from analysis. For comparisons between genes of interest, expression data were normalized to WT mice on the same diet. All assays were performed according to manufacturer protocols.

### 2.7. 16S rRNA Library Preparation and Sequencing

Library preparation and sequencing were performed at the University of Missouri DNA Core Facility. Bacterial 16S rRNA amplicon libraries were generated via amplification of the 16S rRNA gene with primers (U515F/806R) previously developed against the V4 region, flanked by Illumina standard adapter sequences [28,29]. Dual-indexed F and R primers were used in all reactions. Amplification was performed in 50 µL reactions containing 100 ng fecal DNA, F and R primers (0.2 µM each), dNTPs (200 µM each), and Phusion high-fidelity DNA polymerase (1U). PCR parameters were as follows: 98 °C^(3min)^ + [98 °C^(15sec)^ + 50 °C^(30sec)^ + 72 °C^(30sec)^] × 25 cycles + 72 °C^(7min)^. Following PCR, amplicon pools (5 µL/reaction) were combined, mixed, and purified using Axygen^TM^ Axyprep MagPCR clean-up beads at an equal volume of 50 µL of amplicons and incubation at room temperature (RT) for 15 min. Following clean-up, products were washed multiple times with 80% ethanol, resuspended in 32.5 µL EB buffer, incubated for two minutes at RT, and then placed on a magnetic stand for five minutes. Final amplicon pools were evaluated using an Advanced Analytical Fragment Analyzer automated electrophoresis system, quantified using a Qubit 2.0 fluorometer and quant-iT HS dsDNA kits, and diluted according to Illumina’s standard protocol for sequencing on the MiSeq instrument using V2 chemistry kits.

### 2.8. Informatics Analysis of 16S rRNA Sequences

Sequenced DNA was assembled and annotated at the University of Missouri Informatics Research Core. Primers were designed to match the 5′ ends of forward and reverse reads. Using Cutadapt (version 2.6; https://github.com/marcelm/cutadapt; [30]) software, the primer sequence and its reverse complement were removed from the 5′ end of the forward read, along with all bases downstream of the latter. The same approach was applied to the reverse read, with primers in the opposite roles. The 16S rRNA libraries were generated at 25 cycles. Read pairs were rejected if either read failed to match a 5′ primer, using an allowed error-rate of 0.1. The Qiime2 dada2 plugin (version 1.10.0; [31]) was used to denoise, de-replicate, and count amplicon sequence variants (ASVs), based on the following parameters: (1) forward and reverse reads were trimmed to 150 bases, (2) forward and reverse reads with >2 expected errors were discarded, and 3) chimera detection and removal were performed using the “consensus” method. R version 3.5.1 and Biom version 2.1.7 were used in Qiime2. Taxonomies were assigned to trimmed sequences using the Silva.v132 database [32], using the classify-sklearn procedure.

### 2.9. Statistics

Statistical methods and number of mice per group can be found in the corresponding figure legends, with the exception of microbiome analyses, which can be found in ‘Methods’.

### 2.10. Statistical Analysis of Annotated Sequences

Statistical analysis of β-diversity between groups was performed using ¼ root transformed relative amplicon sequence variant (ASV) abundances. Experimental groups were compared with one-way permutational multivariate analysis of variance (PERMANOVA) of Bray-Curtis and Jaccard distances using Past 3.26b. The corrected *p*-values for pairwise statistical comparisons were calculated using Bonferroni’s method. Heatmaps and hierarchical clustering dendrograms were generated in Metaboanalyst 4.0 (https://www.metaboanalyst.ca/). Data were cube-root transformed and clustered using Ward’s method based on Euclidean distances. The top 50 most statistically significant ASV’s were determined by ANOVA.

## 3. Results

### 3.1. Young Male 5xFAD Mice Exhibited Normal Overall Energy Homeostasis When Fed Normal Chow Diet (NCD)

In order to evaluate baseline energy homeostasis in 5xFAD mice compared to WT mice, we used indirect calorimetry for metabolic profiling and measured locomotor activity, energy expenditure, oxygen consumption, nutrient utilization, and food intake in WT and 5xFAD animals fed NCD. In order to minimize the confounding effect of advanced neurodegeneration on energy homeostasis, we used male 5xFAD and control animals at 2 months of age. We determined that WT and 5xFAD mice had comparable body weight and body composition (Figure 1A–C). Then, mice were housed individually in a home cage environment and acclimated for 48 h before recording. 5xFAD mice had a normal diurnal rhythm of locomotor activity during ad libitum feeding (Figure 1D,E). Energy expenditure (EE) and oxygen consumption (VO_2_) were also similar between 5xFAD and WT control mice (Figure 1F–I). Energy partitioning measured by respiratory exchange ratio (RER) was used to gauge carbohydrate and lipid utilization as metabolic fuel. An RER of 1.0 indicates carbohydrate being the predominant fuel source, while a value of 0.7 indicates the combustion of fatty acids as the predominant fuel source. We found no significant differences of RER in WT and 5xFAD mice during ad libitum feeding (Figure 1J,K). WT and 5xFAD mice also had comparable food intake during ad libitum feeding (Figure 1L,M). In order to further evaluate the potential impact on neuroendocrine system for energy balance regulation in the 5xFAD mice, we applied the fasting-refeeding experimental paradigm in the indirect calorimetry experiment. Consistent with the results from ad libitum feeding, we found that the metabolic parameters were virtually indistinguishable between WT and 5xFAD mice during fasting-refeeding challenge (Appendix A). Collectively, our results showed that 5xFAD mice on normal chow diet were able to maintain energy homeostasis with normal food intake, energy expenditure, and nutrient utilization.

### 3.2. 5xFAD Mice Fed Normal Chow Diet Exhibited Age-Dependent Sexual Dimorphic Effects on Body Weight Maintenance

Though younger 5xFAD mice were able to maintain energy balance, advanced neuropathology in older 5xFAD mice could impair the maintenance of whole-body energy homeostasis due to defects in feeding-related locomotion and neurocognitive function. We analyzed 7.5-month-old male and female WT and 5xFAD mice fed on normal chow diet to measure body weight and adiposity. Female 5xFAD mice had lower body weight, whereas male 5xFAD mice were comparable to WT mice (Appendix A). We collected the liver, epididymal white adipose tissue (EWAT), and pancreas from both male and female mice and measured the organ masses and found that female 5xFAD had lower masses of liver, EWAT, and pancreas (Appendix A). When tissue weights were normalized to total body weight, female 5xFAD mice had significantly lower relative EWAT mass (Appendix A). Our results suggested that mechanisms governing energy homeostasis in 5xFAD mice were impaired in an age-dependent sexually dimorphic manner and that female 5xFAD mice on chow diet were more prone to have lower body weight due to adiposity loss.

### 3.3. 5xFAD Mice Displayed Metabolic Defects after High-Fat Diet (HFD) Feeding

In order to evaluate the effect of HFD on energy homeostasis in 5xFAD mice, we switched the diet from NCD comprised of 13.2% calories from fat to HFD comprised of 60% calories from fat for two months and then repeated metabolic profiling. 5xFAD mice exposed to HFD feeding had similar body weight (Figure 2A), lean mass (Figure 2B), and fat mass (Figure 2C) as WT mice. However, 5xFAD mice had greater nocturnal locomotor activity during ad libitum feeding, particularly towards the end of the dark phase when mice typically exhibit declined activity (Figure 2D,E). The increases in locomotor activity were accompanied by increases in energy expenditure and oxygen consumption (VO_2_) in 5xFAD mice during ad libitum feeding (Figure 2F–I). However, we did not observe differences in respiratory exchange ratio (RER), suggesting that energy partitioning was not different between 5xFAD and WT mice (Figure 2J,K). During ad libitum feeding, 5xFAD mice consumed more HFD than WT mice especially during the dark phase (Figure 2L,M, significant after 31.7 h).

As we observed changes in feeding and energy expenditure during ad libitum feeding in HFD-fed 5xFAD mice, we further examined the satiety and energy balance regulation using the fasting-refeeding challenge. In response to food deprivation, 5xFAD mice had significantly increased locomotor activity, indicating increased hunger and food foraging behavior (Appendix A). After refeeding, 5xFAD mice had significantly increased locomotor activity, energy expenditure, and oxygen consumption, indicating increased feeding behavior (Appendix A). Consistent with foraging behaviors, 5xFAD mice consumed more HFD during the 24-h refeeding phase (Appendix A, significant after 20.9 h). Taken together, we concluded that 5xFAD mice on HFD had reduced satiety and increased caloric intake but increased energy expenditure.

### 3.4. 5xFAD Mice on High-Fat Diet (HFD) Have Altered Glycemia and Blood Lipid Profile Compared with WT Mice

We investigated whether increased ad libitum HFD food intake in 5xFAD mice would alter serum metabolites that are involved in metabolic syndrome and also implicated as AD risk factors [33,34,35]. Male 5xFAD mice on HFD had decreased blood glucose concentration during ad libitum feeding but not during fasting or refeeding (Figure 3A). WT and 5xFAD mice had similar concentrations of serum insulin and cholesterol throughout the changes to physiological status (Figure 3B,C). We observed higher serum non-esterified fatty acids (NEFA) during fasting and higher glycerol during refeeding (Figure 3D,E), suggesting 5xFAD mice had increased lipolysis during the fasting-refeeding challenge. During refeeding, we also observed significantly increased serum triglycerides (TG) in 5xFAD mice (Figure 3F). Taken together, we concluded that 5xFAD mice on HFD have increased serum lipid metabolites (e.g., increased NEFA and TG) compared to WT mice, which is consistent with their metabolic defects on HFD (e.g., food intake) (Figure 2).

### 3.5. High-Fat Diet (HFD) Exacerbated the Glucose Intolerance Phenotype in Male 5xFAD Mice

Impaired glucose metabolism and cellular bioenergetics in the brain have been associated with AD pathology [4,5,12]. HFD feeding causes nutritional excess and impairs hypothalamic regulation of energy balance and glucose homeostasis [36,37]. In order to investigate the dietary effect on glucose metabolism in 5xFAD mice, we evaluated the glucose tolerance phenotype using male 5xFAD and control mice fed with NCD and HFD. We first performed an oral glucose tolerance test (oGTT) using young male 5xFAD and WT mice fed NCD with comparable weight and adiposity. 5xFAD mice had a modest but significantly higher peak glucose concentration at 30 min (Figure 4A). After 2 months of HFD feeding, we performed a glucose tolerance test again and found that 5xFAD mice showed decreased glucose clearance during oGTT (Figure 4B). In addition, we measured the body weight throughout the time course of HFD feeding and found that 5xFAD and WT mice fed HFD had similar weight gain (Figure 4C), suggesting that glucose intolerance was not confounded by differences in body weight adiposity gain. Therefore, we concluded that male 5xFAD mice were more susceptible to HFD-induced glucose intolerance.

### 3.6. High-Fat Diet (HFD) Altered the Gut Microbiome Composition in both WT and 5xFAD Mice 

As dietary change causes rapid and profound change in the gut microbiome composition, we investigated the relative contribution of the 5xFAD genetic background and HFD to alterations in the microbiome composition and the possibility of specific interactions between HFD and the 5xFAD background. We measured the relative abundance of bacterial taxa in feces of WT and 5xFAD mice on NCD or HFD by 16S rRNA gene sequencing. In 28 samples, 1404 amplicon sequence variants (ASV’s) were identified with an average read count of 7.3 × 10^4 per sample. The results were then rarefied to 258 ASV’s that were used for downstream analyses to characterize the microbiome composition (Figure 5). We determined that HFD changed the relative abundance in the phyla *Firmicutesi*, *Bacteroidetes* and *Actinobacteria*, with no discernable differences between WT and 5xFAD mice of the same diet (Figure 5A,B).

We also selected the top 50 of the most statistically significant ASV’s and performed hierarchical clustering (HC) based on Euclidean distances. HC showed that the differences in microbial compositions were mainly attributed to the diet, and that dissimilarities between WT and 5xFAD mice were relatively small compared to the dissimilarities between diets (Figure 5C). We further confirmed this result using principle coordinate analysis (PCoA) based on Jaccard and Bray-Curtis distances (Figure 5D,E). In both analyses, the only significant comparisons were between the different diets, which shows that the dissimilarities between samples were mainly driven by diet and not genotype (Figure 5D,E).

Finally, we determined the α-diversity and richness of the gut microbiome in 5xFAD and WT mice fed either NCD or HFD. The richness of the gut microbiome was decreased in HFD samples compared to NCD samples, but no effect of genotype was observed (Figure 5F). We measured α-diversity, which considers the evenness and richness of the microbiome compositions, using the Shannon-H and Simpson-1d indices. Diet caused a significant change in the Shannon-H index and a nearly significant change in the Simpson-1d index, but no genotype effects were observed (Figure 5G,H). Taken together, we concluded that changes in gut microbiome composition were mainly driven by HFD and that 5xFAD and WT mice on the same diet have similar gut microbiome compositions.

### 3.7. Amyloidogenic and Inflammatory Pathways in the Hippocampus of 5xFAD Mice Are Exacerbated by High-Fat Diet (HFD)

We examined the transcripts of genes with known functions in metabolism and AD pathology. First, the mRNA of genes involved in the insulin signaling pathway, such as *Insr*, *Akt3*, and *Pik3r2*, were downregulated in NCD-fed 5xFAD mice but upregulated in HFD-fed 5xFAD mice compared to WT mice (Figure 6A). Of note, *Mtor*, which mediates nutrient sensing and interacts with the insulin signaling pathway, was increased in 5xFAD mice on HFD. Therefore, our data demonstrated increased transcription for key regulators of cellular energy metabolism in 5xFAD mice on HFD, which suggests compensatory response to diet-induced nutrient excess at the transcription level. Secondly, genes genetically associated with AD risk, including *Apoe*, *Lrp1*, *Clu*, *App*, and *Psen2*, had further increased mRNA levels in 5xFAD mice on HFD (Figure 6B). Thirdly, in order to determine whether HFD could exacerbate the neurotoxicity of plaques in 5xFAD mice, we assessed the expression of apoptosis and pro-apoptosis pathway genes in HFD-fed 5xFAD mice and found that many were further upregulated by HFD feeding (Figure 6C). *Casp8*, *Casp6*, *Nfkb1,* and *Atm* showed increased transcription in 5xFAD mice fed on HFD compared to NCD. Lastly, many of the genes with the highest fold change in 5xFAD mice were microglial markers (Figure 6D). Of note, *Cx3cr1* and *Tmem119* were further upregulated in HFD-fed 5xFAD mice, suggesting that HFD feeding increased the expression of microglia-specific marker genes. Taken together, our results demonstrated that HFD feeding promoted pathological progression of AD in 5xFAD hippocampi, possibly due to increased plaque burden, neuronal death, and microglial mediated neuroinflammatory response. In addition to the individual genes with well-established roles in metabolism and AD pathology, we unbiasedly analyzed the differentially expressed genes in 5xFAD mice fed NCD versus HFD based on fold change and statistical significance (Figure 6E). The top hits were plotted in Figure 6F, which provides a list of potential targets to mitigate the effects of HFD in 5xFAD mice.

In order to understand the overall physiological impact of HFD on cellular pathways in the hippocampus of 5xFAD mice, we performed Nanostring pathway analyses for 5xFAD mice on NCD versus HFD (Figure 6G and Appendix A). Consistent with our findings based on analyses of individual gene expression (Figure 6A–D), many of the top differentially regulated pathways were related to insulin signaling, cellular stress, neurotransmission, cytokine signaling, microglial function, and immune response. Furthermore, we performed gene set enrichment analysis (GSEA) with DAVID 6.8 (https://david.ncifcrf.gov/) to identify and rank the pathways that are altered in HFD-fed 5xFAD mice (Figure 6H). The 10 most significant pathways from Kyoto Encyclopedia of Genes and Genomes (KEGG) Pathway and Gene Ontology (GO) terms were selected and ranked by fold-enrichment. Our GSEA results revealed that the top HFD-induced changes to pathways in 5xFAD mice were related to the regulation of neurological function (e.g., synaptic plasticity, neurotransmission, and neuronal death) and metabolic function (e.g., epigenetic and rhythmic regulation, protein phosphorylation, and cAMP signaling).

## 4. Discussion

Epidemiological and clinical studies have highlighted the role of diet and nutrition in the development of AD [18,21]. Given the ethical and technical barriers to conduct mechanistic studies in human subjects, the emerging Alzheimer’s disease animal models present as excellent alternative preclinical models for identifying biomarkers and developing therapeutics in basic and translational research [38]. In the current study, we used 5xFAD mice as an Alzheimer’s disease model to study the physiological and molecular underpinning between diet-induced metabolic defects and AD pathology. Specifically, we set to address two questions–(1) whether 5xFAD mice were more susceptible to high-fat diet (HFD)-induced metabolic disorders; (2) whether HFD could increase stress on AD-related neurological pathways. The dietary fat in HFD was rendered from lard and comprised 60% of the calories. First, we systematically characterized the metabolic parameters of 5xFAD and control mice on normal chow diet (NCD) versus high-fat diet (HFD) using indirect calorimetry. We found that HFD feeding disrupted energy balance in male 5xFAD mice, leading to increased locomotor activity, energy expenditure, and food intake. We further measured the glucose tolerance and circulating lipid metabolites under different feeding statuses. Our results demonstrated that 5xFAD mice had glucose intolerance, which was worsened by HFD feeding. Moreover, high dietary fat intake led to elevated circulating lipids (i.e., TG and NEFA) in 5xFAD mice. We also characterized the gut microbiome composition of the WT and 5xFAD mice fed NCD and HFD. Though we found no taxonomical differences associated with genotype, HFD caused profound changes in the microbiome composition, which could cause altered microbial products and host immune response. Finally, we isolated and quantified hippocampal mRNAs related to AD neuropathology and neuroinflammation and showed that HFD elevated the expression of apoptotic, microglial, and amyloidogenic genes in 5xFAD mice. We performed comprehensive analysis of the cellular pathways in 5xFAD mice fed NCD vs. HFD. Our gene ontology (GO) analysis showed that the differentially expressed genes were enriched in GO terms which included long-term synaptic plasticity, insulin signaling, and neuron death.

In our current studies, 5xFAD mice on HFD displayed increased energy intake and expenditure compared to WT animals (Figure 2). Moreover, 5xFAD mice on HFD displayed more glucose and lipid metabolism defects compared to WT animals (Figure 3 and Figure 4). Therefore, our studies demonstrate that 5xFAD mice were more susceptible to HFD-induced metabolic defects, which could create a vicious cycle of impaired metabolic fitness and cognitive decline. Several mechanisms may contribute to this phenotype and connect metabolic disorders with AD. HFD compromises brain glucose and insulin sensing, which are required for maintaining metabolic homeostasis [39,40]. Increased triglycerides were reported to cross the blood-brain barrier, leading to inhibition of neuronal insulin receptor signaling [41]. HFD causes inflammation in the hypothalamus and impairs energy balance regulation by reducing hypothalamic proopiomelanocortin (POMC) neurons and increasing gliosis [37,42]. Our previous studies have shown that increasing the hormonal sensitivity in POMC neurons ameliorates the metabolic derangements caused by long-term HFD feeding in aged mice [27]. Future study is warranted to understand whether the neuroendocrine system is more severely impacted in 5xFAD fed HFD. Alternatively, HFD could precipitate Aβ deposition and inhibit Aβ degradation. For example, HFD feeding in another AD mouse model (APP23 mice) caused cognitive deficits and increased hippocampal and cortical Aβ deposition [43] and insulin resistance was reported to increase hyperphosphorylation of Tau via GSK-3β activity [44].

We performed the hippocampal gene transcript profiling using NanoString nCounter assays, which enables accurate and higher-throughput transcriptional analysis of gene panels related to AD neuropathology and neuroinflammation. Our key finding was that the transcription of insulin signaling genes, AD risk genes and microglial markers were among the most significantly upregulated genes in HFD-fed 5xFAD hippocampi (Figure 6). (I) We found Insulin signaling pathway genes, including *InsR*, *Akt3*, *Mtor*, and *Prkaa2*, were upregulated in HFD-fed 5xFAD mice compared NCD-fed 5xFAD mice (Figure 6A). mTOR and AMPK are important cellular sensors for energy status and were implicated in AD-related pathologies [6,45,46]. Since HFD increases insulin resistance in the hippocampus, resulting in impaired cognitive performance [47,48,49,50], our findings suggest that these genes were upregulated in order to compensate for insulin resistance. (II) We discovered that the expression of several AD risk-associated genes was further increased in 5xFAD mice fed HFD (Figure 6B). *Apoe* and its receptor, *Lrp1*, were further upregulated in HFD-fed 5xFAD mice, suggesting that HFD caused disturbances of brain cholesterol metabolism. 5xFAD is an amyloid pathology model that expresses human *APP* and *PSEN1* transgenes with a total of five AD-linked mutations. The amyloid plaque burden is mainly driven by the human transgene expression. In our study, we detected mouse *App* and *Psen2* were also upregulated in 5xFAD mice fed HFD, suggesting that HFD may have increased endogenous App generation and App processing. *Clu*, which interacts with extracellular Aβ plaques, was also increased. Therefore, our data suggest that HFD modifies the expression of AD risk-associated genes which are likely to increase amyloid generation in 5xFAD mice. (III) We discovered that HFD further increased transcripts encoding *Casp8* and *Casp6* in 5xFAD mice suggesting that increased apoptosis (Figure 6C). *Nfkb1*, which is a pro-apoptotic gene, was also upregulated. Therefore, we concluded that HFD increased neurotoxicity in 5xFAD hippocampi which may lead to greater apoptosis and neurodegeneration. (IV) Microglial markers, including *Trem2*, *Cd68*, *Cx3cr1*, *Plcg2*, and *Tmem119*, were selectively upregulated with the highest fold changes in 5xFAD mice compared with WT mice (Figure 6D). Interestingly, the expression of *Cx3cr1* and *Tmem119* was further upregulated on HFD, indicating the dietary effects were significantly associated with expression levels of microglia-specific marker genes. (V) We performed differential expression analysis by comparing HFD-fed with NCD-fed 5xFAD mice (Figure 6E,F). The top hits with the most statistical significance are genes involved in insulin signaling and cellular energetics, such as *Insr*, *Akt3*, *Pten*, *Creb1*, *Prkar2a*, *Becn1*, *Atg3*, and *Gsk3b*. (VI) In addition to identifying the differential expression of individual genes, we performed pathway analysis to pinpoint altered molecular and cellular pathways in HFD-fed versus NCD-fed 5xFAD mice (Figure 6G,H and Appendix A). Of note, innate and adaptive immune response, cytokine signaling, microglia function, activated microglia, and inflammatory signaling pathways had high undirected global significance (>5), suggesting that HFD feeding caused greater inflammatory response and microglial activation in the hippocampi of 5xFAD mice (Figure 6G). Our findings in HFD-fed 5xFAD mice were consistent with previous reports that HFD increases microglial activation in the hippocampus [51,52,53] which may promote synapse loss [51]. Collectively, our data showed that HFD combined with amyloid pathology is likely to increase stress on multiple pathways and cause detrimental effects on long-term synaptic plasticity, neuronal apoptosis, and neuroinflammation.

Multiple experimental and clinical studies have pointed out that changes in gut microbiome composition contribute to the progression of metabolic and neurodegeneration diseases via altered microbial metabolites, immune activation, and bacterial amyloids [54,55,56,57]. The role of the gut microbiome in neurodegenerative diseases is beginning to be elucidated. The gut microbiome is a source of bacterial amyloids (which can cross-seed with Aβ), endotoxins (i.e., lipopolysaccharides), and inflammatory cytokines which can prime immune cells in the brain [54,55,56,57]. Treating APP/PS1 mice with antibiotics reduced plaque formation, suggesting that the microbiome secretes factors which accelerate disease pathogenesis [58]. Studies in human patients found correlations between Aβ and phospho-Tau peptides in the cerebrospinal fluid and specific genera of microbes [57]. Interestingly, transplantation of HFD-associated microbiomes increased anxiety behaviors and impaired auditory-cued fear learning compared to NCD-transplanted microbiomes [59]. However, it is unclear whether the different microbiome compositions in AD patients are directly causal to impaired cognitive function. In our study, the effects of 5xFAD on the microbiome were negligible compared to the effect of HFD feeding (Figure 5), suggesting that diet-induced changes to the gut microbiome are more rapid and of greater magnitude than changes induced by AD pathology.

A previous study reported that HFD enhances cerebral amyloid angiopathy and cognitive impairment in 5xFAD mice independently of metabolic disorders [24]. The major difference between that study and ours was the age difference when 5xFAD mice were used for metabolic analyses. When 13 month-old 5xFAD mice were switched to HFD for another 10 weeks, they gained less weight and glucose tolerance was not affected compared to WT mice [24]. 5xFAD mice demonstrate age-dependent rapid progression to neurodegeneration [26]. Therefore, the reduced body mass and weight gain of older 5xFAD mice could be caused by the advanced state of neurodegeneration. Moreover, lower body weight of aged 5xFAD mice could be a confounding factor for the glucose tolerance tests. In order to minimize these confounding factors associated with the older 5xFAD mice, we used younger mice for metabolic profiling. Male 5xFAD mice on either NCD or HFD had similar body weight and body composition as the WT mice and were used in our studies. Meanwhile, another group used younger 5xFAD mice to start HFD feeding and found increased amyloid deposition and defects in glucose metabolism [25], which was consistent with our findings in the current study. We found NCD-fed female (but not male) 5xFAD mice at 7.5 months of age had reduced body weight and adiposity (Appendix A), indicating an association between cognitive decline and frailty [60]. Differences in body composition would be a confounding factor for characterizing the glucose and energy metabolic phenotype in female mice, therefore we focused the current study on male mice. How female 5xFAD mice would respond to dietary changes in terms of metabolism and neuropathology is an interesting question and will be investigated in future studies.

## 5. Conclusions

In conclusion, our studies demonstrated that 5xFAD mice were more susceptible to HFD-induced metabolic disorders and that HFD could exacerbate stress on AD-related neuropathological and neuroinflammatory pathways. Altogether, our data suggest that targeting metabolic dysfunctions caused by high dietary fat intake can ameliorate AD symptoms via effects on insulin signaling, neuroinflammation, Aβ deposition, and microglia activation in the hippocampus.

## Figures and Tables

**Figure 1 nutrients-12-02977-f001:**
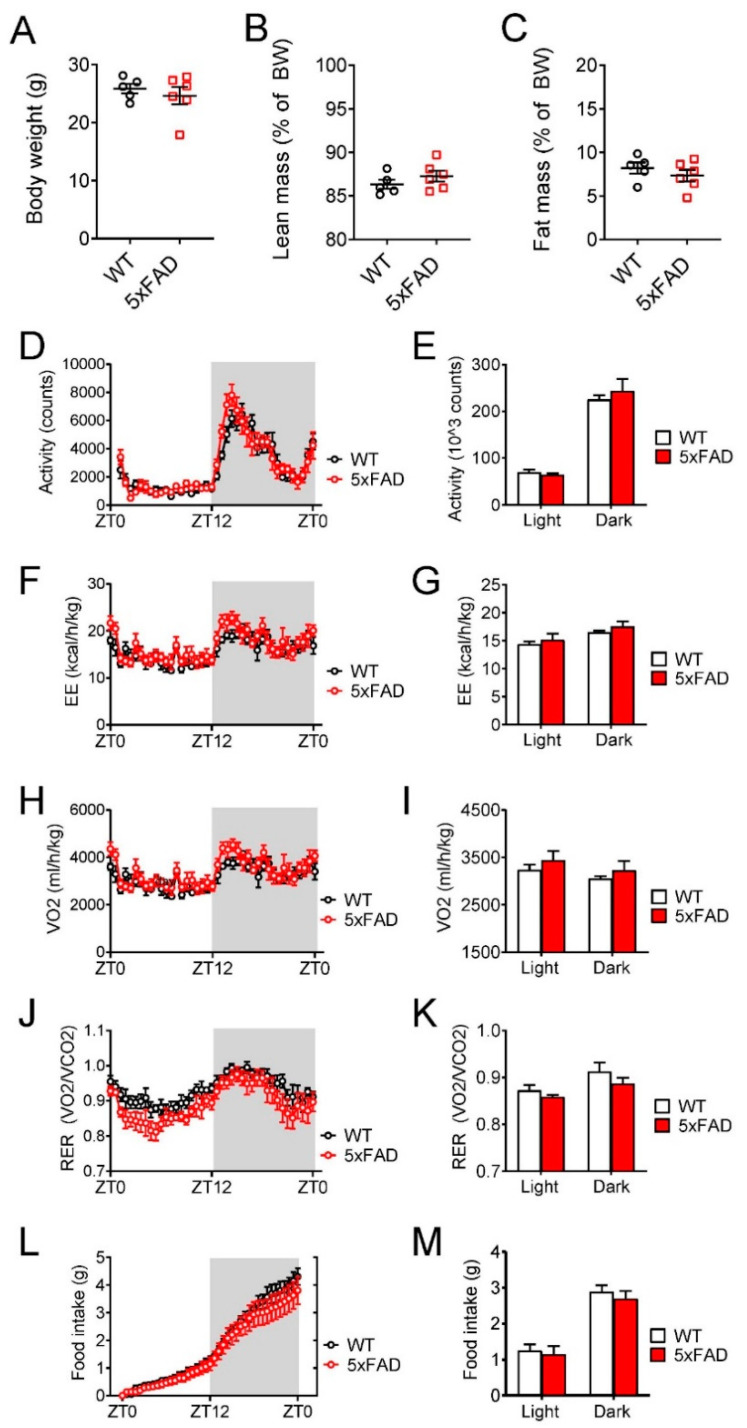
Young male 5xFAD mice fed normal chow diet (NCD) had normal energy homeostasis. Male 5xFAD and wild type (WT) mice were 2 months old and fed normal chow diet (NCD). Metabolic profiling was performed during ad libitum feeding. (**A**) Body weight. (**B**) Lean mass as a percentage of body weight (BW). (**C**) Fat mass as a percentage of body weight. (**D**) Lateral locomotor activity and (**E**) total locomotor activity. (**F**) Energy expenditure (EE) normalized to total body weight and (**G**) average EE normalized to total body weight. (**H**) Oxygen consumption (VO_2_) and (**I**) average VO_2_. (**J**) Respiratory exchange ratio (RER) and (**K**) average RER. (**L**) Cumulative food intake and (**M**) total food intake during each light/dark phase. Data are displayed as means ± standard error of the mean. Statistical comparisons for time-course data were calculated with Fisher’s least squared difference method. Statistical comparisons for light/dark averages were calculated with student’s *t*-test. *n* = 5 for WT mice and *n* = 6 for 5xFAD mice.

**Figure 2 nutrients-12-02977-f002:**
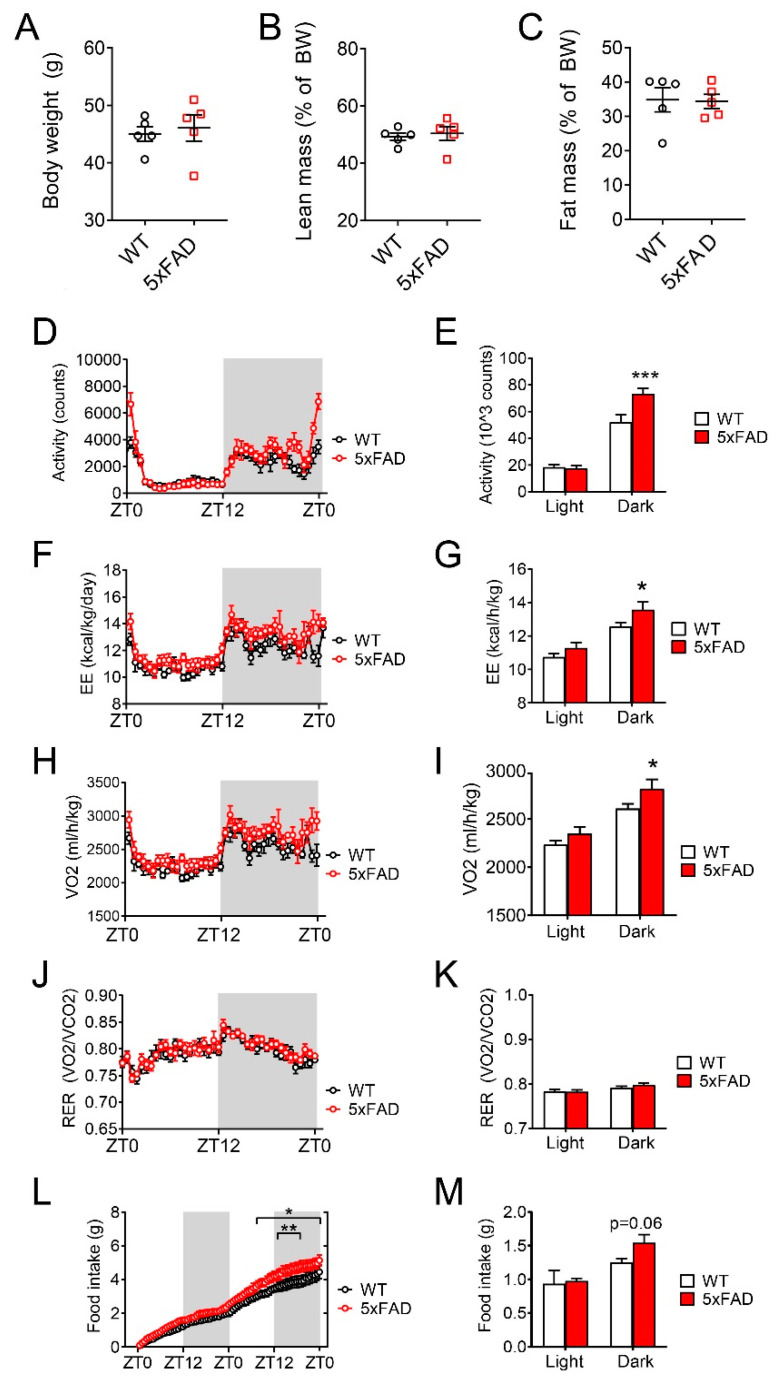
5xFAD male mice displayed metabolic defects after high-fat diet (HFD) feeding. Male 5xFAD and wild type (WT) mice were fed HFD for 2 months. Metabolic profiling was performed during ad libitum feeding on HFD. (**A**) Body weight. (**B**) Lean mass as a percentage of body weight (BW). (**C**) Fat mass as a percentage of body weight (BW). (**D**) Lateral locomotor activity and (**E**) total locomotor activity. (**F**) Energy expenditure (EE) normalized to total body weight and (**G**) average EE normalized to total body weight. (**H**) Oxygen consumption (VO_2_) and (**I**) average VO_2_. (**J**) Respiratory exchange ratio (RER) and (**K**) average RER. (**L**) Cumulative food intake and (**M**) total food intake during each light/dark phase. Data are displayed as means ± standard error of the mean. Statistical comparisons for time-course data were calculated with Fisher’s least squared difference method. Statistical comparisons for light/dark averages were calculated with student’s *t*-test. Statistical comparisons for bar graphs were calculated with student’s *t*-test. (*) indicates *p* < 0.05; (**) indicates *p* < 0.01; (***) indicates *p* < 0.001. *n* = 5 for each group.

**Figure 3 nutrients-12-02977-f003:**
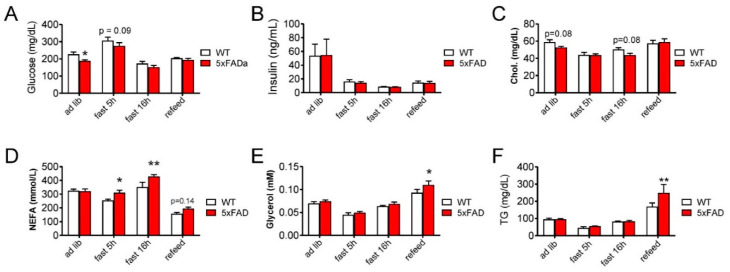
5xFAD male mice on high-fat diet (HFD) had altered blood glucose and lipid metabolites compared with wild type (WT) mice. Serum metabolites were measured in 6-month-old mice fed HFD for 3 months. (**A**) blood glucose, (**B**) serum insulin, (**C**) serum free cholesterol, (**D**) serum non-esterified fatty acids (NEFA), (**E**) serum glycerol, (**F**) serum triglycerides (TG). Data are displayed as means ± standard error of the mean. *n* = 5 for WT mice and *n* = 6 for 5xFAD mice. Statistical comparisons were calculated with student’s *t*-test, * *p* < 0.05, ** *p* < 0.01; additional *p*-values were noted in the panels for clarity.

**Figure 4 nutrients-12-02977-f004:**
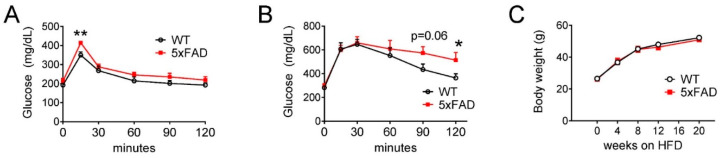
High-fat diet (HFD) exacerbated the glucose intolerance phenotype in male 5xFAD mice. (**A**) Glucose excursion of 2-month-old male mice fed NCD during oral glucose tolerance test (oGTT) (2 g/kg). (**B**) Glucose excursion during oGTT (3 g/kg) in wild type (WT) and 5xFAD male mice fed HFD for 2 months. (**C**) Body weight of WT and 5xFAD mice on HFD. Data are displayed as means ± standard error of the mean. *n* = 5 per group. Statistical comparisons for each timepoint were calculated with student’s *t*-test, (*) indicates *p* < 0.05, (**) indicates *p* < 0.01.

**Figure 5 nutrients-12-02977-f005:**
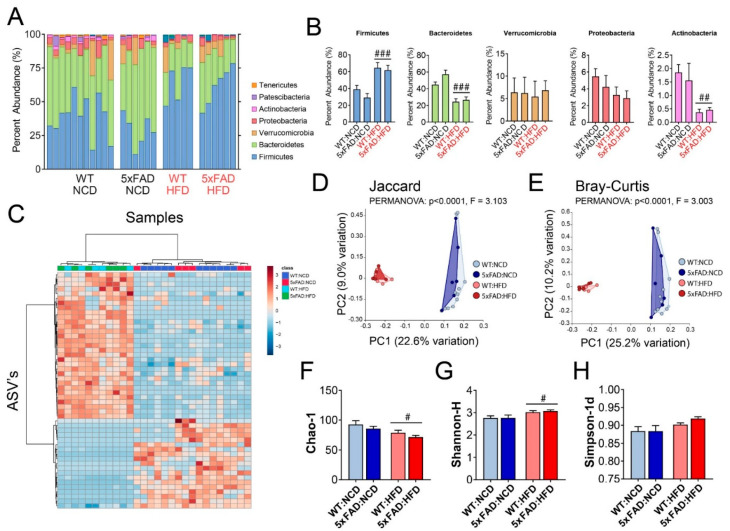
High-fat diet (HFD) altered the gut microbiome composition in both WT and 5xFAD mice. The microbiome compositions and diversity in fecal samples from 6-month-old wild type (WT) and 5xFAD mice fed normal chow diet (NCD) or high-fat diet (HFD) were determined by 16S rRNA sequencing. (**A**) Stacked bar chart of the relative abundance of detected bacterial phyla. Each stacked bar represents an individual mouse with genotypes indicated on the x-axis. Low abundance phyla <1% were omitted from the legend. (**B**) Average relative abundance of detected bacterial phyla: *Firmicutes*, *Bacteroidetes*, *Verrucomicrobia*, *Proteobacteria*, and *Actinobacteria*. (**C**) Hierarchical clustering heatmap of the top 50 significant amplicon sequence variants (ASV’s). Columns represent individual mice with respective genotypes indicated by the legend. (**D**) Principle coordinate analysis (PCoA) of Jaccard distances. (**E**) Principle coordinate analysis (PCoA) of Bray-Curtis distances. (**F**) Richness measurement using Chao-1 index. (**G**) α-diversity measurement using Shannon-H index. (**H**) α-diversity measurement using Simpson-1d index. Statistical comparisons in PCoA were made using one-way permutational multivariate analysis of variance (PERMANOVA). Bonferroni-corrected pairwise comparisons between genotypes on the same diet were not statistically significant. Statistical comparisons in B, F-H were made using two-way ANOVA, where (#) indicates *p* < 0.05 between different diets, (##) indicates *p* < 0.01, and (###) indicates *p* < 0.001 between different diets. Tukey-corrected pairwise comparisons between genotypes on the same diet were not statistically significant. *n* = 11, 6, 5, 6 mice per group for WT:NCD, 5xFAD:NCD, WT:HFD, and 5xFAD:HFD, respectively.

**Figure 6 nutrients-12-02977-f006:**
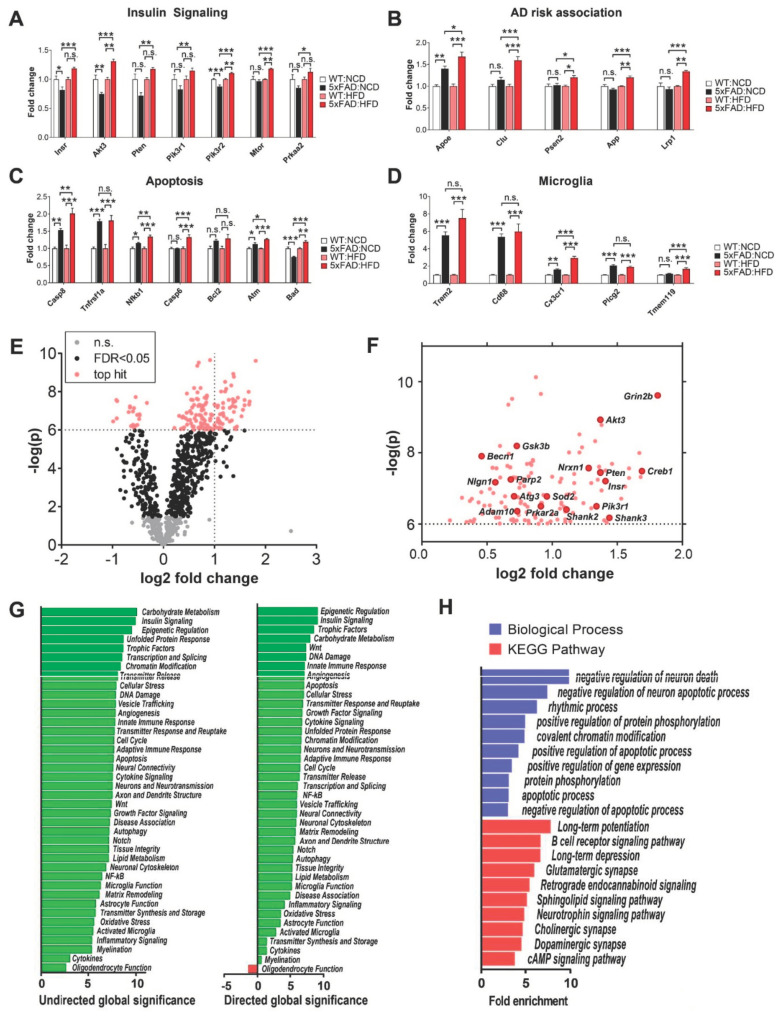
Amyloidogenic and inflammatory pathways in the hippocampus of 5xFAD mice were exacerbated by high-fat diet (HFD). RNA profiling was performed using samples isolated from the hippocampi of wild type (WT) and 5xFAD mice fed NCD and HFD. (**A**) NanoString gene expression analysis of insulin signaling genes, (**B**) AD risk-associated genes, (**C**) apoptosis-related genes, (**D**) microglia associated genes. (**E**) Volcano plot of hippocampal transcripts from 5xFAD mice fed HFD versus NCD. Top hits are genes with −log10(*p*) > 6. Out of a total 979 gene probes, 302 were non-significant and 677 gene probes were significant at Benjamini-Hochberg false-discovery rate (FDR) < 0.05. (**F**) Labeled genes of interest that were top hits identified in Figure 6E (**G**) Nanostring undirected global significance (left) and directed global significance (right). Undirected global significance is a measure of differential expression of genes belonging to each pathway, calculated as the square root of the mean squared t-statistics. Similarly, directed global significance measures the tendency to have up or down regulated genes, calculated as the square root of the mean of signed (positive/negative) squared t-statistics. (**H**) Gene set enrichment analysis (GSEA) of significant hippocampal mRNAs (FDR < 0.05) from 5xFAD mice fed NCD versus HFD. The top 10 KEGG pathways and Gene Ontology (GO) term biological processes were identified based on *p*-value, then sorted by highest fold-enrichment. Data are displayed as means ± standard error of the mean. *n* = 6,6,5,6 for WT:NCD, 5xFAD:NCD, WT:HFD, and 5xFAD:HFD, respectively. Statistical comparisons between groups in A-D were performed using two-way ANOVA and Tukey post-hoc tests. (*) indicates *p* < 0.05, (**) indicates *p* < 0.01. (***) indicates *p* < 0.001. Comparisons that were non-significant (n.s.) are also indicated for clarity.

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
