# Peer review of "Metabolic Defects Caused by High-Fat Diet Modify Disease Risk through Inflammatory and Amyloidogenic Pathways in a Mouse Model of Alzheimer’s Disease"

_nutrients, 2020, doi:10.3390/nu12102977_

Round 1

Reviewer 1 Report

The paper investigates the effect of HFD or chow control on metabolism and AD-related gene expression in WT and a mouse model of Alzheimer's disease. Overall, it is an interesting paper but more detailed descriptions of the methods are needed as detailed below. Additionally, some of the graphs are unclear as specified (mainly Figures 1 and 2).

It is unclear why the effect of diet per genotype is not investigated. The one analysis were both diet groups are described at the same age, the analysis of gut microbiome, even states that the changes were mainly driven by diet rather than genotype. Additionally, the paper very strongly focuses on the effects on male mice. Only one section mentions female mice (body weight maintenance), without seeming to explain why females are suddenly investigated too.

[Lines 64-65] Would you expect a Mediterranean diet to have protective effects? May be interesting to do alongside the chow/HFD comparison.

[Lines 88-91] Were the HFD preserved the same way? Why were HFD and chow control mice euthanised differently; surely that would differentially affect some of their biochemistry?

[Line 94] Why weren't the HFD raised on the same chow diet?

[Lines 102-104] Why were the HFD and chow control mice fasted for different time points, and given different proportions of glucose (2 vs 3 g/kg). What was the glucose vehicle? Was it given via gavage? How long did the GTT last? At what time points was blood glucose measured? Were any other blood parameters measured, e.g. insulin?

[Serum biochemistry] Were all analyses performed at all 4 fasting regimes (fed, 5hr, 16hr, re-fed)? At what ages were these analyses, and OGTTs EchoMRI and indirect calorimetry, undertaken?

[Lines 114-117] Were the indirect calorimetry measurements only taken for 24hr (one light/dark cycle)?

[Line 120] Which side was the hippocampus taken from?

[Lines 120-122] Why was the RNA extracted using different methods per diet group? Was the HFD RNA purified in the same way as the chow control group?

[Lines 139-140] Please use units for the time specifications.

[16S rRNA Library prep and sequencing] Please report manufacturers for all kits/reagents/instruments used.

[Line 147] Which MiSeq reagent kit did you use? How many cycles were run?

[Lines 182-183 and 115-116] In the results the indirect calorimetry is stated to measure locomotor activity, energy expenditure, oxygen consumption, nutrient utilisation, and food intake, while in the Methods only food intake, energy expenditure and RQ are listed as being measured every 36 min. At what intervals are the other measures taken?

[Lines 184-186] This description of information about the chosen mouse model may be more helpful earlier on, perhaps in the introduction.

[Figures 1-3] Was this data recorded for female mice as well?

[Figure 1] Some of the bar graphs look close to significance. Is it possible to include the individual data points as shown in Figures 1A-C?

[Figures 1-4 and 6] Was a power calculation done to confirm that only 5-6 mice per group was sufficient?

[Figures 1 and 2] The line graphs need to be reformatted, perhaps with smaller data points, so that both groups can be seen - Figures 1L and 2J are particularly obscured.

[Lines 224-226] Again, this descriptive information may be more useful in the introduction.

[Figures 1,2 and 4] Figure 1 appears to reflect mice at 2 months of age, whereas Figure 2 is mice that were fed a third diet for 3 months and then HFD for 2 months, making them 5 months of age. Why did you not conduct these experiments at the same age point so the effect of diet could be examined for each genotype?

[Figure 4] Please add graph C to the legend.

[Figures 5 and 6] Is the in male mice?

[Figures 5 and S2] What were the sample sizes?

[figure 6] At what age were the RNA samples collected?

[Lines 361-362] Wouldn't the different ages (2 vs 5 months) make a direct comparison of control diet and HFD inaccurate?

[Lines 457-459] As stated in the results, the RNA analysis was to investigate if HFD exacerbated AD-related pathways, with the finding that "HFD feeding promoted AD pathological progression in 5xFAD hippocampi" (Lines 328-329). More explanation is needed as to how this equates to non-HFD ameliorating AD symptoms, as opposed to simply not making them worse.

Author Response

"Please see the attachment".

Reviewer 2 Report

Strengths:
- The studies in this work show that a mouse model of Alzheimer's disease (5xFAD) was more susceptible to metabolic disorders induced by high fat diets (HFD) and that this could exacerbate stress in neuropathological disease related to Alzheimer's disease and neuroinflammatory pathways.
- In addition, they have found positive regulation of genes of the insulin signaling pathway and metabolism in mice fed HFD,
- Several genes associated with the risk of Alzheimer's disease associated with the generation (increase) of beta-amyloid plaques were also up-regulated by the action of HFD,
- On the other hand, they also discovered that HFD increased the expression of apoptotic genes and neurodegeneration due to neurotoxicity.
- HFD positively regulated a greater inflammatory response and microglial activation in the hippocampus of these 5xFAD mice, promoting the loss of synapses and their plasticity.
- Lastly, the effects of the microbiome were negligible compared to the effects of the HFD diet.

Questions / Weaknesses:
- Lines 90-91: For what reason are control animals with a normal diet sacrificed under anesthesia and HFD mice with CO2?

1) Line 96: Please transfer “lard” from line 360 ​​to experimental procedures. One of the doubts was what type of lipid source or fatty acid composition (saturated: palmitic, lauric and / or stearic) was used in the experimentation.

2) In the Glicose measurements section (Lines: 100 - 104): I understand the differences in the composition of carbohydrates of the two types of diets, being much higher in NCD, but why not perform the oral GGT test for both groups with the same amount (2g / kg) after fasting at the same fixed time?

3) Section Hippocampal mRNA quantitation (line 119): please explain why you extract RNA with trizol reagent for samples from mice on HFD diet, and yet for NCD samples you use STAT-GO reagent. Which is the reason?

4) Line 237 (for figure 2L): Please, could you indicate in the text the time-course data where these differences occurred (specify time)? The same for Supplemental figure 3.

5) Line 240-241: “….In response to food deprivation, 5xFAD mice had significantly increased locomotor activity, indicating increased hunger and food foraging behavior….”.

Increased hunger or could it be from hedonic control? Are you aware or have you ever measured enkephalins or circulating neuropeptide Y?

6) Figure 2L: you draw a 48-hour timing on the X axis, which, unlike the rest (for example, Figure 1L) only reflects 24 hours. Why doesn't it represent the other 24 hours? Furthermore, during the first 24 hours of Figure 2L, it can be seen that the nocturnal intake with HFD in the two groups of animals is much lower than with the NCD control diet.

7)- Figure 3: The letters D, E and F do not correspond to those in the figure.
- Figure 4: you do not indicate the description of the letter C.
- Figures 5 and 6 are cut on the right side by the margin. Try to adjust.
- In addition, figure 6 does not have a footnote or legend that indicates that it is figure 6.

8) Line 315: Did you want to state that 5xFAD was positively regulated in mice on the HFD diet?

On the other hand, it is true that the significant difference is made by the type of diet, but it seems that the transgenic itself has the ability to upregulate gene expression. Except for the regulation of the Mtor gene, in addition to the HFD diet, it seems that with the NCD control diet there is also gene overexpression of these non-significant genes in 5xFAD.

Reviewer 3 Report

The aim of this study was to highlight the physiological and molecular underpinning between diet-induced metabolic defects and Alzheimer’s disease pathology. This is an interesting study and it represents a potentially valuable addition to the literature focusing on the interplay between metabolic and inflammatory  status and the neuropathological progression in AD mouse model.  

The manuscript reports interesting observations; it is well-structured, comprehensive and clear presented; the methodology and analyses provided are both accurate and properly conducted.

Below few suggestions to the authors:

  • Line 191: please check the references to the figures
  • In the “experimental procedures” paragraph (line95) was indicated :”At 3 months old mice were switched to HFD”; in the figure legend 1 was reported “Male 5xFAD and WT” were 2 months old. In the figure legend 2 please specify the age of the animals. The time points of the experimental design are confusing.
  • Lines 184-186 and lines 280-282 could be moved in the discussion section

The aim of this study was to highlight the physiological and molecular underpinning between diet-induced metabolic defects and Alzheimer’s disease pathology. This is an interesting study and it represents a potentially valuable addition to the literature focusing on the interplay between metabolic and inflammatory  status and the neuropathological progression in AD mouse model.  

The manuscript reports interesting observations; it is well-structured, comprehensive and clear presented; the methodology and analyses provided are both accurate and properly conducted.

Below few suggestions to the authors:

  • Line 191: please check the references to the figures
  • In the “experimental procedures” paragraph (line95) was indicated :”At 3 months old mice were switched to HFD”; in the figure legend 1 was reported “Male 5xFAD and WT” were 2 months old. In the figure legend 2 please specify the age of the animals. The time points of the experimental design are confusing.
  • Lines 184-186 and lines 280-282 could be moved in the discussion section
